# Pre-Intervention Effects of a Community-Based Intervention Targeting Alcohol Use (LEF); The Role of Participatory Research and Publicity

**DOI:** 10.3390/ijerph18168823

**Published:** 2021-08-21

**Authors:** Ina M. Koning, Vincent G. Van der Rijst, John B. F. De Wit, Charlotte De Kock

**Affiliations:** 1Youth Studies, Interdisciplinary Social Science, Utrecht University, P.O. Box 80140, 3508 TC Utrecht, The Netherlands; v.g.vanderrijst@uu.nl; 2Social Policy and Public Health, Interdisciplinary Social Science, Utrecht University, P.O. Box 80140, 3508 TC Utrecht, The Netherlands; j.dewit@uu.nl; 3Institute for Social Drug Research, Faculty of Law and Criminology, Ghent University, B-9000 Ghent, Belgium; charlotte.dekock@ugent.be

**Keywords:** participatory research, pre-intervention, community-based intervention, alcohol

## Abstract

This study explores the impact of the ‘pre-intervention effects’ of a community-based intervention. This refers to participatory research processes and parallel publicity in the media on changes in alcohol use and relevant mechanisms (rules and norms about alcohol, accessibility of alcohol in a formal setting) among adolescents before any intervention is implemented. The aim was to investigate the contribution of these processes (i.e., pre-intervention effects) to changes in intervention-targeted factors before any actual intervention was implemented. In a quasi-experimental study, data were collected twice by means of self-report among adolescents living in two municipalities (control and experimental condition). A regression analysis showed negative pre-intervention main effects on adolescents’ perceived accessibility of alcohol in a formal setting. Moreover, among adolescents aged 15 years and older, the normative decline in strictness of rules and norms was less steep in the experimental condition compared to the control condition. Additionally, adolescents aged 14 years and younger in the experimental condition reported more weekly drinking compared to their peers in the control condition. No differential effects across gender were found. To conclude, applying a co-creational approach in the development of an intervention not only contributes to more effective interventions in the end, but the involvement of and discussions in the community when planning the intervention contribute to changes in targeted factors. This implies that public discussions about the development of intervention strategies should be considered as an essential feature of co-creation in community-based interventions.

## 1. Introduction

Most people have their first experience with alcohol during adolescence. Several studies have shown that in recent years, the number of underage adolescents that drink alcohol has declined on a global scale [1,2], including in The Netherlands [3,4]. For example, among Dutch 12 year olds, the prevalence of lifetime and monthly drinking decreased from 71.1 and 30.9 percent in 2003 to 20.2 and 3.7 percent in 2017, respectively [4]. This decline is most likely the result of increased awareness of the harmful effect of alcohol on adolescent development, as well as the imperative role of parents as targets in prevention efforts [5]. Although alcohol use among adolescents has declined, the mean age for the onset of drinking is still relatively young: adolescents in The Netherlands are on average 13.3 years old when they start drinking alcohol [3]. Moreover, once they start, adolescents drink at high levels: of those who have drunk alcohol in the past month, 71% have been involved in binge drinking (drinking at least five glasses of alcohol during an occasion) [3,4]. Early-onset alcohol use is an important risk factor for many negative consequences, like the development of other substance (ab) use [6], co-morbid mental health problems [7], more aggressive behavior [8], a lower school engagement, and poorer educational outcomes later in life [9]; therefore, prevention of early alcohol use in adolescents is crucial.

The development and implementation of interventions on alcohol use in adolescents are complex, as risk factors are multi-faceted, involving a combination of individual, social and environmental factors [10], with drinking behaviors influenced by factors from different levels, i.e., individual factors, parental and peer influence, and alcohol policies [11]. Moreover, the use of alcohol is often considered a social activity, done together with peers [12,13]. Consequently, several studies showed that interventions are more effective in reducing adolescents’ alcohol initiation and use when they are not only targeting the adolescents themselves [14], but also their parents [15,16], and their entire family [17], or school [18]. Thus, considering that factors at multiple levels are relevant, multi-component interventions are more effective than single-component interventions and can therefore better contribute to reducing adolescents’ alcohol use.

Community-based interventions are designed to address the complex interplay between individual behaviors and broader societal influences by involving the community itself and may be an effective way of reducing alcohol use and related harms. Traditional interventions that mainly focus on components at a community level (i.e., community-level interventions) include interventions such as warning and informing about the negative consequences of alcohol use. However, community-based interventions also aim to involve (part of) the community. They can subsequently influence perceptions about social norms and can add three different types of intervention strategies: regulatory (e.g., strengthening alcohol policy enforcement), physical (e.g., reducing the visibility of alcohol in public), and economic (e.g., making alcoholic beverages more expensive) strategies [19]. Thus, the aim of community-based interventions is not to intervene on an individual level alone but to bring stakeholders like policymakers, educators, and regulators, together to influence this broader environment in which drinking frequently occurs [20].

Ensuring the appropriateness of community-based interventions to the local context requires the participation of local community actors (e.g., councilors, educators, parents, youth) who have intimate knowledge of the local context. Rationales for community participation in intervention development can be roughly split into two potentially complementary models: the utilitarian model and the empowerment model [21]. A utilitarian model of community participation posits that involvement of the community is essential because it will improve the effectiveness and sustainability of the resulting intervention due to it being better in line with the local context and having greater buy-in from local actors who were involved in the development. An empowerment model argues that community participation is an end in itself as it will directly contribute to community empowerment [22,23], which aligns with the vision of health promotion as set out in the Ottawa Charter (World Health Organization; [24]).

Reviews of community-based interventions using participatory and engaged research have identified various program-related outcomes, such as an impact on communities’ health and social outcomes, community-wide behavioral and environmental changes, community empowerment, and reductions in social inequalities [25,26]. For example, once engaged, youth involved in interventions show a decrease in substance use [27]. These outcomes provide evidence to support a utilitarian model of community participation to improve health and social outcomes. These are promising outcomes, but uncoupling the outcomes from the intervention remains a challenge, and so does contributing the participatory research processes and its public discussions to the outcomes. This is largely due to evaluating interventions only at the end of an intervention. Moreover, the evaluation often reduces the definition of the intervention of the program that was delivered to the research participants rather than encompassing the whole process from co-creation to evaluation. In order to evaluate the co-creation process, a baseline assessment should be conducted at the start of this process, followed by another assessment before the actual interventions are implemented, most ideally in an experimental trial. A recent realist review by Jagosh et al. [28] unpacked an array of outcomes that could be directly attributed to the participatory research processes, including capacity building, culturally appropriate research and subsequent adaptation, productive conflict resolution, and systems change. Whilst this review provides a good starting point to unpack the benefits of participatory research more broadly, it leaves the question of how participatory research processes directly improves health and social outcomes unanswered.

A community-based intervention, referred to as LEF (Dutch translation of ‘courage’), applied a co-creational approach to the development of the intervention strategies. This period of development of intervention strategies took about six months and received much attention in the media, e.g., via interviews in (online) local newspapers and discussion in open civic meetings. The studied outcomes (early drinking) as well as the factors contributing to it—particularly the accessibility of alcohol—were a topic of discussion in the media, as well as among members of the community. Due to the fact that the Principal Investigator (first author) of this study is part of the community, she was often (in)directly called upon by inhabitants about these topics of discussion. This led to the question of if the increased discussions and attention towards the drinking behaviors among youth in the development of the intervention may have affected both the outcomes themselves and the mechanisms, without any actual intervention strategy already implemented. We refer to these effects as the ‘pre-intervention’ effects, i.e., the effects of the development of an intervention while using co-creation, on relevant outcomes before intervention strategies are implemented. It is likely that this pre-intervention phase already has an impact on the targeted mechanisms and outcomes of interest.

This study aims to address this gap by focusing on the process of participatory development of a community-based intervention aiming to reduce alcohol use amongst adolescents. The “intervention” for the context of the current study is the participatory research that was undertaken during the development of the community-based intervention LEF. It explores whether “pre-intervention effects” exist, referring to changes in alcohol use and relevant mechanisms amongst adolescents following the participatory research processes and parallel publicity in the media to develop the intervention before the intervention is implemented. These ‘pre-intervention’ effects were examined across gender and age groups as it is known that boys and older adolescents are more likely to drink alcohol [4] and may therefore be less affected. This study will give insight into a potentially relevant phase in intervention research which may form the basis for enabling any change and therefore may impact the level of change in outcomes. The main aim of this study is to test to what extent these pre-intervention effects exist and contribute to change in targeted outcomes.

## 2. Methods

### 2.1. Procedure

The LEF program is a community-based intervention in the municipality of Edam-Volendam in The Netherlands that aims to delay the onset of alcohol use and reduce weekly drinking among youth. A quasi-experimental design is used including, two municipalities: one experimental municipality and one control municipality. In February 2018, both public secondary schools within the municipality were willing to participate in the experimental condition, while another public secondary school within the fairly similar municipality of Enkhuizen participated in the control condition. In the experimental municipality as well as the control municipality, there are two schools that could participate. However, one of two schools in the experimental condition did not fully participate as students were not allowed to fill out the questionnaire during regular school hours. Consequently, only one school was selected in the control condition, based on the best match with the school that fully participated in the experimental condition. The risk of spill-over effects is highly unlikely as the control school is situated more than 43 km away. Though the experimental municipality is twice the size of the control municipality (36,000 vs. 18,500 inhabitants respectively), in many other ways, the municipalities are comparable in terms of percentage of 12–18 year old youths (9% vs. 8%), 18+ alcohol users (29% vs. 32%) and 18+ excessive alcohol users (7% in both municipalities).

Data were collected by trained research assistants in classrooms using online questionnaires available on a secured website. Parents received a letter of consent, which informed them about the participation of the school in the program, and they were given the opportunity to refuse participation of their child (1.13% refusal). Data were gathered in May/June 2018 (TN), and again six months later in November/December 2018 (T_0_) before any actual intervention was implemented. At TN, the aim was to conduct a needs assessment in which an explanatory model was designed (see Figure 1), while T0 functioned as the formal baseline assessment. The study was approved by the Ethics Review Board of the Faculty of Behavioral & Social Sciences at Utrecht University (FETC18-060).

### 2.2. Intervention: Participatory Development of a Community-Based Intervention

The LEF program is a community-based intervention developed, implemented, and evaluated in a municipality in The Netherlands. The developmental phase of LEF consisted of three steps (see Figure 2).

First, a needs assessment was conducted, including semi-structured interviews with all relevant stakeholders (e.g., local politicians, parents, adolescents, youth, heads of school). In addition to these interviews, self-report questionnaires among the youth were conducted. By collecting local data, the LEF intervention components could be directly matched to the local target group and context to increase the likelihood of effectively changing the desired outcomes [29,30]. Focusing on youth/parent engagement in this phase can also overcome potential obstacles perceived by youth to participate and thereby increase the relevance of and the participation of youth/parents when implementing the intervention [31]. Based on the results of both data sources, the outcomes of the intervention were defined by the researcher and the municipality; delay the onset of drinking among underage adolescents.

Second, this data, alongside (inter)national knowledge about important determinants of the onset of drinking, were used to identify factors that were relevant for early drinking in this specific community [32]. This resulted in an explanatory model (Figure 1) including important and changeable factors within three domains, i.e., (a) parents (e.g., rules about alcohol), (b) youth (e.g., norms about alcohol), and (c) repression (e.g., lower accessibility of alcohol).

Third, by mobilizing a local ‘task force’ (a group that was differently composed every meeting), input from members of the community was gathered (e.g., through discussion meetings per domain) to translate these relevant factors into intervention strategies [19]. In addition to inviting specific stakeholders directly to join the task force, participation in this task force was open to anyone interested. This co-creation approach, together with existing local intervention strategies, theories of behavior change, and other scientific knowledge, was used to develop the first intervention strategies targeting factors in the explanatory model.

Following the needs assessment, the social norms about drinking among adolescents, rule-setting by parents, and accessibility of alcohol, emerged as the most important and changeable mechanisms that can influence the onset of alcohol use of adolescents.

### 2.3. Sample

All secondary schools and all classes except for the exam classes (classes at the end of their study) participated in the study. A total of 2893 students were asked to participate in the study. Of these, 524 students did not participate due to their parents’ refusal or their absence from school on the day of data collection (individual or whole class due to scheduling problems). In addition, students of the second school were only allowed to fill out the questionnaire outside school hours in their own time; 63 students out of 286 participated. This resulted in a sample eligible for analysis of *N* = 2146.

Socio-demographic characteristics and variables of interest for each condition and the total sample are presented in Table 1. The total sample had a mean age of 14.67 (*SD* = 1.33), consisting of 48.1% boys. At baseline, the experimental condition differs significantly from the control condition in terms of more positive norms, less strict rules about alcohol, easier access to alcohol, and more weekly drinking (Table 1). At T_0_, adolescents in the experimental condition reported significantly easier access to alcohol and more weekly drinking than adolescents in the control condition.

### 2.4. Loss to Follow-Up

Participants who did not participate in the follow-up (*n* = 311; 14%) differed from completers in terms of perceived accessibility of alcohol outside home (*t* = 2.28, *p* = 0.023), age (*t* = 3.58, *p* < 0.00) and level of education (*t* = 4.20, *p* < 0.00). That is, adolescents who did not participate in the follow-up reported higher accessibility of alcohol outside the home, were on average somewhat younger, and were more likely to be in a lower level of education.

### 2.5. Measures

The main outcomes measured were related to the mechanisms of and final envisaged outcome of the LEF program, i.e., norms about youth drinking, rules about alcohol, perceived accessibility of alcohol (home and café), and weekly drinking.

*Norms about youth drinking* reflect the acceptability of adolescents consuming alcohol in various situations (for example at home, and at a party with friends). Participants were asked to what degree they thought it is acceptable for a person of the same age to drink alcohol in various situations (1 = ‘not acceptable at all’, to 5 = ‘very acceptable’). The instrument is based on a Dutch translation of the ‘Alcohol Use Norms Scale’ [33], which was used in earlier research in the Netherlands [34]. Originally it contained seven items; in this study, we used five items. A mean score was calculated, where a higher score indicated more positive norms about youth drinking. Cronbach’s alpha was 0.92 at both waves.

*Rules about alcohol* reflect the degree of parental rule-setting regarding alcohol use as experienced by the adolescent [35]. Participants were asked to what extent their parents approve they (would) drink alcohol in various situations (for example, at home with parents present and at a party with friends). Originally it contained ten items; in this study, we used five items, which were scored on a 5-point scale (1 = ‘never’ to 5 = ‘always’). All items were recorded, and a mean score was calculated, where a higher score indicated stricter parental rules about alcohol use. Cronbach’s alphas were 0.94 (TN) and 0.92 (T_0_).

*Perceived accessibility of alcohol* reflects the degree of the perceived ease of obtaining alcohol in various situations. Participants were asked how easy they (think they would) get alcohol in various situations on a 5-point scale (1 = ‘very easy’ to 5 = ‘impossible’). Perceived accessibility is divided in *formal* (at a pub/café) and *informal* (mean score of three situations: at home from their parents, at a friend’s home, and at the home of relatives; Chronbach’s alphas were 0.78 (TN) and 0.75 (T_0_)) accessibility of alcohol.

*Weekly drinking* was measured by using the quantity-frequency measure [36,37]. Frequency was measured by asking the number of days the adolescents usually drink on a weekly basis, while quantity was measured by asking how many glasses of alcohol the adolescents usually drink on a typical day they drink alcohol (9-point Likert scale; 0–6 = I do not drink alcohol to 6 glasses per day, 8 = 7–10 glasses and 9 = 11 glasses or more). The quantity-frequency was computed by calculating the products of the number of days and the number of glasses, where higher scores indicated more weekly drinking.

Age was measured by asking the age of the participant in years. The legal age for buying alcoholic drinks in The Netherlands was set to 18 relatively recently (January 2014.) In the experimental municipality, this change in legal age is enforced to a limited extent. Age 15, one year for the former legal drinking age, is considered by many parents as an appropriate age for their children to start experimenting with alcohol. This is exemplified by the steep increase in the prevalence of drinkers from age 14 onwards [32]. Therefore, age was dichotomized into two categories (0 = 14 years or younger and 1 = 15 years and older).

### 2.6. Data Analysis

Missing data were estimated in Mplus using the Full Information Maximum Likelihood Estimation with Robust Standard Errors default setting (MLR), allowing information of all the 2164 participants to be used for analysis [38]. Descriptive statistics and correlations were retrieved for the total group, for the experimental condition and control condition separately, and for each measurement occasion. *T*-tests were conducted to assess differences in all variables of interest between the control and experimental condition at baseline and follow-up. We used a Multiple Regression analysis to test the impact of the pre-intervention process as applied in LEF on the mechanisms and outcomes at T_0_, i.e., *norms about youth drinking, rules about alcohol, accessibility of alcohol (informal and formal), and weekly drinking*. The direct effects were examined for the dichotomous intervention variable (0 = control, 1= experimental) at baseline on the dependent variables at the follow-up while controlling for the outcomes variable at baseline, as well as age and gender. Since *weekly drinking* has a high variance relative to the mean, which indicates overdispersion, a zero-inflated negative binomial model was used [39]. The MLR estimator is robust to the non-normality of the variables *norms about youth drinking* and *rules about alcohol* and is, therefore, used in the analysis. In line with Nieminen, Lehtiniemi, Vähäkangas, Huusko, and Rautio [40], we used the standardized βs as effect size indices, whereby β < 0.2 was considered small, 0.2 < β < 0.5 moderate, and β > 0.5 a strong effect. To test moderation by age and gender, the interaction terms between the intervention condition × age (centered) and intervention condition × gender were added to the model separately.

## 3. Results

### 3.1. Main Effects

The multiple regression analysis of the experimental condition at TN on the mechanisms at T_0_ (*norms about youth drinking, rules about alcohol, informal and formal accessibility of alcohol*) and outcomes (*weekly drinking*) while controlling for the outcomes at TN show two significant effects (see Table 2). That is, adolescents in the experimental condition are significantly more likely to report a lower ease of access to alcohol in formal settings compared to adolescents in the control condition (*β* = −0.05, *p* = 0.04). Moreover, adolescents in the experimental condition are more likely to have a higher average weekly drinking level than adolescents in the control condition (*β* = 0.28, *p* < 0.00). No other significant main effects of the intervention condition on mechanisms were found.

### 3.2. Moderation Effects

The interaction between condition and gender revealed no significant interaction effects. This means that the effect of the LEF pre-intervention on the mechanisms and outcomes do not significantly differ across gender. All but one (formal accessibility on condition*age) interaction of condition with age showed a significant effect on the mechanisms and outcomes. That is, 15+ aged adolescents in the experimental condition are more likely to report less positive norms about youth’ drinking (*β* = −0.21, *p* < 0.00) and stricter rules about alcohol (*β* = 0.16, *p* < 0.00; Figure 3) compared to 15+ aged adolescents in the control condition. In addition, for older adolescents in the experimental condition, a lower ease of access to alcohol in the home situation was reported compared to adolescents in the control condition (*β* = 0.16, *p* < 0.001). Last, the positive main effect of condition on weekly drinking only applies to adolescents aged 14 and younger (*β* = −0.91, *p* < 0.00).

## 4. Discussion

The current study demonstrated that applying a co-creational process and its subsequent discussions in the community (media) in the development of a complex community-based intervention yielded effects on outcomes and mechanisms. That is, the main effects of pre-intervention were found on adolescents’ perceived accessibility of alcohol in a formal setting. Moreover, among adolescents aged 15 years and older, the normative decline in strictness of rules and norms was less steep in the experimental condition compared to the control condition. Moreover, adolescents aged 14 years and younger in the experimental condition reported more weekly drinking compared to their peers in the control condition.

Although it is recognized that adolescence is typically known for its increase in involvement in risk behaviors, such as the use of alcohol [2,3], alongside the development of more positive norms and norms towards these behaviors when it becomes more salient [41], the current study demonstrated that this normative increase in alcohol use and its norms could be altered by applying a co-creation process where the public is involved in the development of an intervention and made visible through publicity in local media. A search in the online repository of a regional newspaper (Noordhollandsdagblad.nl) on the keywords ‘alcohol and (name municipality)’ shows that in the municipality of Edam-Volendam, three times as many newspaper articles were published in the year of the program (14 December 2017–13 December 2018), compared to 1.5 times as many in the control municipality Enkhuizen. This increased media attention was an unintended consequence, emerging from the three steps in intervention development. Most likely, the increased attention in the media and public discussions about alcohol use among youth may have contributed to these changes in norms and alcohol use. As suggested by social norms theories, e.g., [42,43], (perceived) norms are likely to affect behavior. Particularly within this particular community, drinking alcohol among youth is considered the norm, which makes it hard for adolescents to deviate from this norm by not drinking alcohol [44]. In their review on the role of social norms in anti-smoking campaigns, ref. [45] argue that exposure to mass media messages “can directly encourage individuals to question existing norms and adopt new ones and can indirectly reduce social acceptability of smoking through public discussion” (p. 180). In this line, the increased attention in the media and public discussion about the drinking behavior of youth within the community, as was observed in LEF, could have caused 15+ adolescents to evaluate and adjust their norms about alcohol use. When adolescents are convinced that not all of their peers drink alcohol and drinking alcohol at their age is not socially accepted, this may have changed their norms about alcohol. The same principle may also have applied to parents. Social comparison theory [46] states that people evaluate themselves by comparing themselves with others, which implies that parents assess to what extent their parenting is adequate based on what other parents say and do [46]. If parents believe that other parents hold more permissive norms towards alcohol use of their children, they can experience pressure to conform to this perceived norm, and therefore become more lenient in their own parenting [47,48], for example, by setting less strict rules about alcohol. The increased attention in the media and public discussion may have led to more interpersonal communication among parents about their parenting behaviors [49], and through these conservations, they got the opportunity to get more accurate perceptions about the norms regarding these behaviors [50]. When parents, through these conservations, learned that other parents, in fact, are less tolerant about their children’s alcohol use than they thought, they can be strengthened in their own opinions and therefore can be empowered to set stricter rules themselves.

This might also have applied to other parental behaviors like the perceived reduction in the accessibility of alcohol at home. The home can be an important source of alcohol for youth [51], and multiple studies have shown that access to alcohol in the home and obtaining alcohol from parents can be linked to increased alcohol use, heavy episodic drinking, and a higher risk of alcohol-related harm (for a review, see [52]). Although these negative effects are well known, parents still “give their children alcohol to teach them how to drink responsibly and to prevent risky drinking with peers” ([53], p. 2). The increased attention in the media, public discussion, and interpersonal communication with other parents might have caused parents to question and adjust such views, resulting in reducing the accessibility of alcohol for their children in the informal context, i.e., at home.

Not only the accessibility of alcohol at home but also the commercial (formal) accessibility of alcohol is a strong predictor of alcohol use [54]. Despite that, it is well known that a strict alcohol policy can help reduce the youth’ access to alcohol [55]; however, compliance with these alcohol policies in The Netherlands is still low. A study by Van Hoof and Gosselt [56] showed that 100% percent of underaged mystery shoppers succeed in buying alcohol at commercial places. Furthermore, twenty percent of 16 year olds who drink alcohol declared they bought and consumed alcohol within hospitality services (e.g., pub or nightclub) at least once a month [3], which indicates that in The Netherlands, alcohol is easily accessible for underage youth. The compliance to an alcohol policy reflects broader societal drinking norms: communities with more conservative drinking norms may be more likely to enact and enforce comprehensive policies [55]. When it is socially accepted that underage youth drink alcohol, there will be less rigorous enforcement on underage drinking. However, the increased attention and public discussion may have altered this norm and thereby made it easier to enforce the alcohol policy more rigorously. In addition, by increasing awareness and visibility of enforcement activities, youth might be deterred from buying and consuming alcohol in public places [55]. All in all, in the developmental phase of an intervention, where co-creation is applied and public discussions take place, the co-creation process can be considered as a pre-intervention that may alter some mechanisms of change before specific intervention strategies are being implemented.

Interestingly, the impact of the pre-intervention effects differs for younger and older adolescents. That is, the less steep decline in strictness of rules and norms was mostly changed among 15+ year-old adolescents in the intervention condition, an age group most likely to be involved in drinking already [2]. Higher involvement in drinking behavior may make the discussion about this topic more applicable to this group which is, therefore, most likely to contribute to a change in norms and rules about alcohol. The lower applicability of the ongoing discussion about alcohol use among younger adolescents may also explain the impact of the experimental condition on more drinking among younger adolescents; their norms and rules were not changed by the intervention. Moreover, in the process of co-creation, most media attention and discussions among community members were given to the restrictive component of the intervention (accessibility of alcohol). As the restriction of accessibility to alcohol in the formal setting mostly applies to older adolescents (15 years and older), this may have contributed to the higher levels of drinking among particularly the younger group. Furthermore, it is also likely that because the younger group did not feel engaged by discussions around alcohol use and availability, they might have even started to oppose it, as a rebellion against changing norms and their parents is developmentally normative [57]. Additional follow-up studies may provide more insight into how drinking behavior further develops in this group. The finding that the most favorable effects are found among 15+ adolescents underlines the importance of the applicability of messages for the target group.

Participatory research in community-based interventions is a complex process undertaken in a complex system. This study highlighted the concept of emergence within complex interventions, which means that components of the interventions will work together to create new and unexpected outcomes. That is, in the participatory development of LEF, the media attention that was generated following the start of the research was an unexpected outcome that led to a change in the context within which the resulting LEF intervention will be implemented. This media attention contributed to an increase in conversations about alcohol use in the community and likely contributed to a change in norms among the adolescents surveyed. This study, therefore, highlights that intervening in a complex system (i.e., community level interventions) might lead to unintended consequences and different outcomes than expected. This emphasizes the need for evaluation approaches that are timely and appropriate to capture these kinds of outcomes [58,59].

### Strengths and Limitations

To our knowledge, this study is the first that revealed the importance of co-creation and publicity (i.e., pre-intervention effects) in community-based interventions to change in outcomes and mechanisms. Though the study has several strengths, such as the quasi-experimental design with longitudinal data, the sample size, and innovativeness of the topic, several limitations should be discussed.

First, though it is highly likely that publicity in the co-creational process contributed to the changes in outcomes, this cannot be concluded for certain. It is possible that including community members in the development of interventions may also influence targeted factors even without public discussions. Future research should disentangle these aspects in the planning process of interventions.

Second, the extent to which the results of this study can be generalized to other contexts (municipalities or countries) is limited [60], and even more so since the municipality of Edam-Volendam is a relatively close community, in which public discussions may be more important to foster change. The context wherein the intervention takes place should be taken into account [61]. However, specifically in a municipality like Edam-Volendam where conservative norms about youth drinking are adhered to, the need for community involvement can be made more visible in such communities.

Third, although loss to follow-up was mostly random due to absence on the day of the study or other activities outside school, it is important to consider differential participation in the interpretation of the current findings.

Fourth, we assumed that the control municipality is similar to the experimental municipality. However, studies investigating community-based interventions use the community as the unity of intervention which increases the likelihood of having a comparison group that is less alike than assumed [60]. Studies including multiple municipalities in each condition and matching municipalities across conditions may be relevant to increase the likelihood of comparing similar groups of municipalities.

## 5. Conclusions

This study offers preliminary evidence that relevant changes in outcomes and mechanisms can already be achieved by discussing the issue at hand and ways to deal with this with those involved. Previous research has demonstrated that involving the community in the entire process of intervention (from development to implementation and evaluation) more likely results in interventions matching the needs of the population within the community and thereby achieve more effectiveness [25,26,27]. In the current study, we showed that applying a co-creational approach in the development of an intervention not only contributes to more effective interventions in the end but that discussions in the community when planning the intervention contribute to changes in targeted factors at the pre-intervention stage. This implies that public discussions about the development of intervention strategies should be considered as an essential feature of co-creation in community-based interventions. Without limiting a preventive intervention to media-only strategies, future community-based studies should consider media attention to share information about the progress and aims of the to-be-developed intervention and strive for co-creation. This may be even more so for community-based interventions targeting topics where social norms are highly relevant, such as substance use, were influencing the broader environment has a significant impact.

## Figures and Tables

**Figure 1 ijerph-18-08823-f001:**
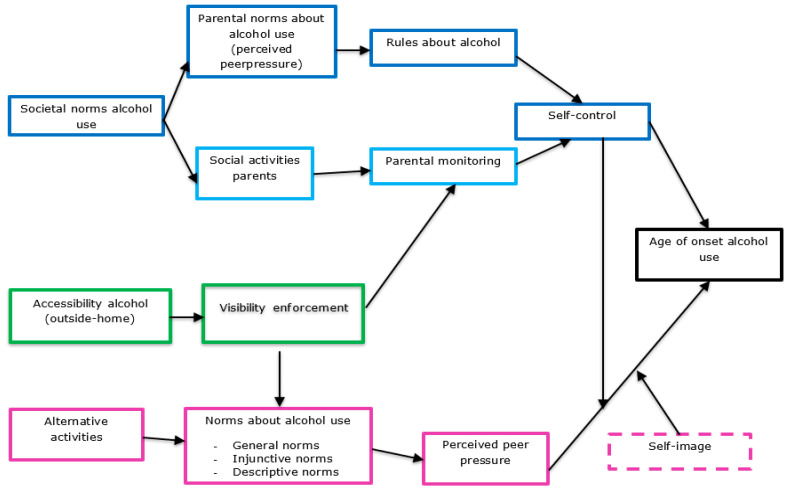
Explanatory Model including Important and Changeable Determinants of Behavior for Onset of Alcohol Use among Youth.

**Figure 2 ijerph-18-08823-f002:**
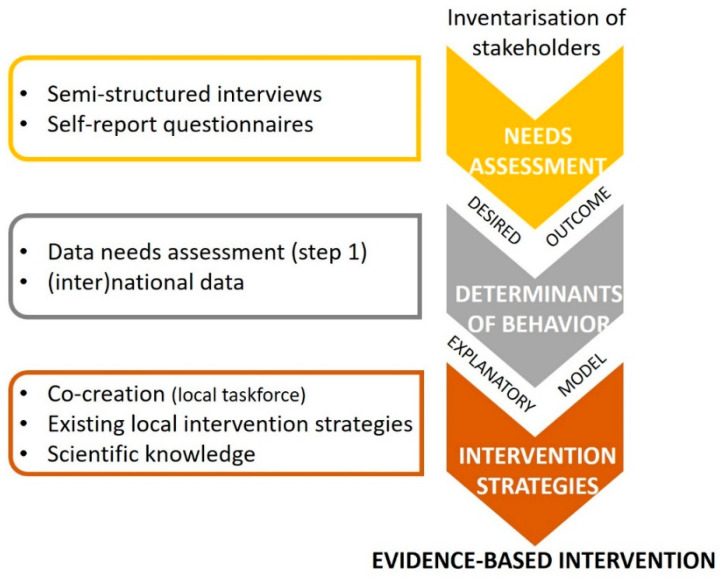
Three Steps Included in the Developmental Phase of the Community-Based Intervention LEF.

**Figure 3 ijerph-18-08823-f003:**
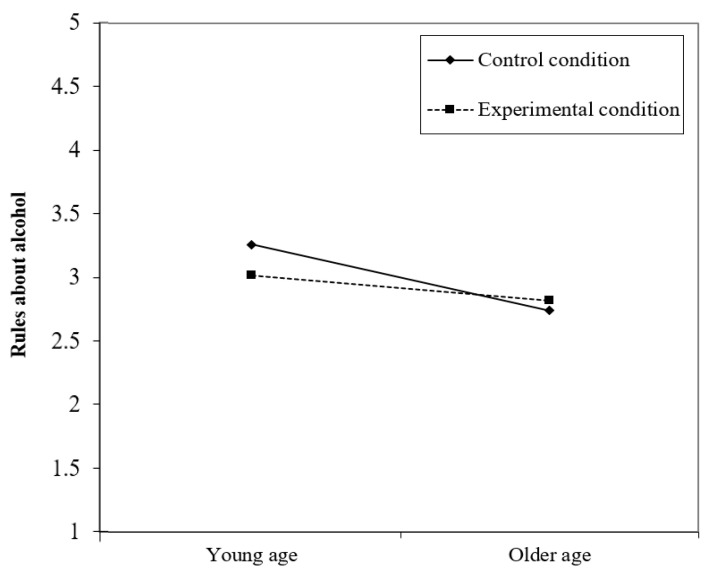
Visualization of the Interaction between Condition*Age on Rules about Alcohol.

**Table 1 ijerph-18-08823-t001:** Descriptive Data on all Variables of Interest for the Control and Experimental Condition and Total Sample.

	Control Condition (*N* = 1027)	Experimental Condition (*N* = 1137)	Total (*N* = 2146)
	*M* (*SD*)	*M* (*SD*)	*M* (*SD*)
TN variables			
Age (M,SD)	14.57 (1.34)	14.75 (1.32) *	14.67 (1.33)
Gender (% boys)	49.9%	46.2%	
Norms about alcohol in youth T_n_ (M,SD)	2.00 (1.14)	2.22 (1.28) *	2.12 (1.19)
Rules about alcohol T_n_ (M,SD)	3.90 (1.29)	3.66 (1.41) *	3.78 (1.36)
Informal accessibility of alcohol T_n_ (M,SD)	3.20 (1.21)	2.97 (1.34) *	3.08 (1.29)
Formal accessibility of alcohol T_n_ (M,SD)	3.84 (1.36)	3.48 (1.53) *	3.65 (1.46)
Weekly drinking T_n_ (M,SD)	1.98 (5.17)	3.61 (8.37) *	2.86 (7.13)
T_0_ variables			
Norms about alcohol in youth T_0_ (M,SD)	1.93 (1.06)	2.04 (1.14)	1.99 (1.10)
Rules about alcohol T_0_ (M,SD)	4.05 (1.17)	3.93 (1.23)	3.99 (1.19)
Informal accessibility of alcohol T_0_ (M,SD)	3.12 (1.13)	2.94 (1.25) *	3.03 (1.19)
Formal accessibility of alcohol T_0_ (M,SD)	3.78 (1.30)	3.48 (1.47) *	3.63 (1.39)
Weekly drinking T_0_ (M,SD)	1.42 (4.96)	2.14 (4.97) *	1.79 (4.97)

* = significantly different from the control condition based on *t*-tests (Chi-square test for gender).

**Table 2 ijerph-18-08823-t002:** Effects of Intervention Condition on Mechanisms (Norms about Youth, Rules about Alcohol, Accessibility of Alcohol) and Outcome (Weekly Drinking) at Follow-Up.

Variable	Norms about Youth Drinking T_1_		Rules about Alcohol T_1_		Informal Accessibility Alcohol T_1_		Formal Accessibility Alcohol T_1_		Weekly Drinking T_1_	
	*β(SE)*	*p*-Value	*β(SE)*	*p*-Value	*β(SE)*	*p*-Value	*β(SE)*	*p*-Value	*β(SE)*	*p*-Value
Age	0.37(0.04)	0.00	−0.33(0.03)	0.00	−0.27(0.03)	0.00	−0.21(0.00)	0.00	0.68(0.09)	0.00
Gender (0 = girls, 1 = boy)	−0.05(0.02)	0.03	0.02(0.02)	0.23	−0.02(0.02)	0.46	0.00(0.99)	0.99	−0.04(0.08)	0.57
Outcome T_0_	0.43(0.03)	0.00	0.59(0.03)	0.00	0.39(0.03)	0.00	0.39(0.02)	0.00	0.65(0.10)	0.00
Condition (1 = experiment)	−0.02(0.02)	0.30	0.02(0.02)	0.43	−0.03(0.02)	0.20	**−0.05(0.04)**	**0.04**	**0.28(0.07)**	**0.00**

Bold values denote statistical significance at the *p* < 0.05 level.

## Data Availability

Not applicable.

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
