# Peer review of "Pre-Intervention Effects of a Community-Based Intervention Targeting Alcohol Use (LEF); The Role of Participatory Research and Publicity"

_ijerph, 2021, doi:10.3390/ijerph18168823_

Round 1
Reviewer 1 Report
- In the introduction, authors said "among Dutch 12-year olds the prevalence of lifetime and monthly drinking decreased from respectively 71.1 and 30.9 percent in 2003 to 20.2 and 3.7 percent in 2017 [4]." It would be interesting to discuss this further. How is this explained?
- In paragraph 6, line 7 of the 1. section, the authors introduce an acronym (i.e., "IP"). Can they clarify what this refers to?
- In paragraph 2, line 5 of the 2.1 section, the authors said "Data were gathered in May/June 2018 (TN), and again six months later in November/December 2018 (T0)". I am concern about the effect of the different periods of time. Could there have been any effects of the school period? If so, what might they be? These may influence the results themselves without being due to the pre-intervention effects. Can authors discuss a little more about these possibilities ?
- I think that the table 1 can be improved to facilitated the reading. For exemple, a second line could be adding under both condition with the Time (T0 and T1) corresponding to the data.
- In section 2.4, authors wrote that "Participants who did not participate in the follow-up (N =311; 14%) differed from completers in terms of perceived accessibility of alcohol outside home (t=2.28, p=.023), age (t=3.58, p<.00) and level of education (t=4.20, p<.00).". It would be interesting to discuss of those results in the discussion since this is relevant for the futur intervention programs.
- In paragraphe 5, line 6 of the 2.5 section authors whore "(7-point Likert scale; 0=I don’t drink alcohol – 11=40 glasses or more)." There is a problem with the 7 point Likert scale and the last response choice corresponding to 11.
- In paragraphe 6 of the 2.5 section, authors said that "Age was measured by asking the age of the participant in years. The measure was dichotomized into two categories; 0=14 years or younger and 1=15 years and older." I wonder why age was dichotomized rather than looking at the variable continuously. This way of classifying the participants may have an effect on the results. For example, young people over the age of 14 are more likely to consume larger quantities of alcohol. Regardless of your decision to dichotomize or not, it would be appropriate to explain this choice.
Reviewer 2 Report
Abstract
- Check grammar of first sentence ‘a community based intervention’
- A sentence is needed for the study aims, before you go into methodology – what is this paper looking to understand?
- In this abstract you say that media was considered a variable that led to changes, but later in the paper you mention that media is an effect of participatory approaches. The role of the media in this paper is a little confusing – are you defining it as a PART of your ‘pre-intervention effects’ which is causing changes in behaviour, or are you considering it as an effect of the participatory approaches?
- “Regression analysis showed preintervention main effects on adolescents’ perceived accessibility of alcohol in a formal setting” – be explicit about what the effect was
- It is not clear what ‘adolescent data’ is. You’ve mentioned you’re collecting from adolescents in the same sentence, what data was collected?
Introduction
- “Reviews of participatory and engaged research in community-based interventions have identified various program-related outcomes, such as an impact on communities’ health and social outcomes, community-wide behavioral and environmental changes, community empowerment, and reductions in social inequalities”. How have these reviews associated the participatory approaches to the outcomes? Or are you saying that community-based programs have these outcomes? Or are you saying that participatory processes + community-based programs have these outcomes? Please clarify.
- You can also write about the evaluation methodologies used to make previous conclusions about the effectiveness of participatory approaches in other studies, and what kind of analysis/methodology is required to make causal links between these approaches and outcomes
- First time using abbreviations, please provide full forms, e.g. PI
- ‘…she was often (in)directly called upon the current issues of the program’ – please clarify this sentence, it’s not clear what you mean.
- “A search in the online catalogues of a regional newspaper (Noordhollandsdagblad.nl) on the terms ‘alcohol [name municipality]’ shows that in the municipality of Edam-Volendam, in the year of the program (14/12/2017-13/12/2018) three times as many newspaper articles were published compared to a 1,5 times increase in the control municipality Enkhuizen. This increased attention in the media was an unintended consequence, emerging from the three steps in intervention development which wasn’t foreseen by the researchers”. This backstory about your research aims is not required here, it can be included in the discussion as further evidence of the effect of participatory approaches. The scientific rationale of the gap in research around the impacts of participatory approaches is enough for your introduction.
Methods
- From the control municipality, only one school participated. Is there only one school in this municipality, or did the other school(s) refuse? Please clarify.
- Were there any important differences between the municipality that may have affected results?
- Under 2.3 Sample – “expect” should be except. Please do a careful proofread of the manuscript.
- Please make an explicit statement before Table 1 on whether the control and experimental groups were similar, and if they differed on any important variable
Discussion
- Please be careful about use of ‘effect’ and ‘affect’. First sentence should be ‘affect’
- The finding that ‘adolescents in the experimental condition are more likely to have a higher average week drinking level than adolescents in the control condition (β=0.28, p<.00)’ should be better addressed in the discussion, as this is a surprising finding. More explicit consideration of this effect found is required, with possible explanations.
